Journal of Data-centric Machine Learning Research (2025)          Submitted 7/24; Revised 1/25; Published 1/25

# Towards impactful challenges: post-challenge paper, benchmarks and other dissemination actions

**Antoine Marot**                                               ANTOINE.MAROT@RTE-FRANCE.COM
*RTE AI Lab, Paris, France*

**David Rousseau**                                               ROUSSEAU@IJCLAB.IN2P3.FR
*Université Paris-Saclay, CNRS/IN2P3, IJCLab, 91405 Orsay, France*

**Zhen (Zach) Xu**                                               ZACH.XU@UCHICAGO.EDU
*University of Chicago, USA*

**Reviewed on OpenReview:** *https://openreview.net/forum?id=AAqyNe12di*

**Editor:** Sebastian Schelter

## Abstract

The conclusion of an AI challenge is not the end of its lifecycle; ensuring a long-lasting impact requires meticulous post-challenge activities. The long-lasting impact also needs to be organised. This chapter covers the various activities after the challenge is formally finished. This work identifies target audiences for post-challenge initiatives and outlines methods for collecting and organizing challenge outputs. The multiple outputs of the challenge are listed, along with the means to collect them. The central part of the chapter is a template for a typical post-challenge paper, including possible graphs and advice on how to turn the challenge into a long-lasting benchmark.

**Keywords:** post-challenge, analysis, paper, dissemination

## 1 Introduction

If winners are announced at the end of a competition, and everyone simply returns to their usual activity, even the best-conceived competition would have been of limited use. Indeed, many AI/ML challenges shine briefly and are then forgotten, leading to missed opportunities for long-term impact. This chapter discusses the various post-challenge activities necessary to ensure lasting effects from such events, some of which should be prepared even before the challenge starts.

AI/ML challenges have become increasingly popular over the last decade, engaging not only researchers but also enthusiasts and industry practitioners. According to the public data from platforms such as MLContests,[1] which gathers information of challenges across platforms, thousands of competitions solving real-world or academia problems are hosted each year. Conferences like NeurIPS, and KDD have also introduced official competition tracks in recent years[2][3], aiming to foster collaboration between experts and provide tangible benchmarks for new methodologies. Blog posts from conference organizers have explained their rationale for hosting more challenges, typi-

---

1. https://mlcontests.com/
2. https://neurips.cc/Conferences/2024/CompetitionTrack
3. https://kdd.org/kdd-cup

cally highlighting their value in addressing open problems, engaging communities, and producing state-of-the-art solutions[4].

Four types of stakeholders typically play specific roles in a challenge: the *organisers*, the *application domain experts*, the *AI/Machine Learning experts* on techniques relevant to the problem posed, and the *participants*. Participants themselves come from diverse backgrounds—ranging from domain-specific experts and AI specialists to students and experienced data scientists—each contributing unique perspectives that enrich the challenge outcomes. Input from all these people should be gathered and made available for posterity so that the community can **build upon results and lessons**, **identify remaining gaps/research directions**, and **access new materials** (benchmarks, codes, tutorials) to further the state of the art.

## 1.1 Why Post-Challenge Activities Matter

Ensuring long-term community engagement requires sharing outcomes and resources with both participants and the wider research community:

- Results and lessons on which they can build upon,

- Remaining gaps and research directions that can push the boundaries of current knowledge,

- Materials (benchmarks, solution codes, visualizations, tutorials) that facilitate continued and reproducible research.

Post-challenge activity is therefore necessary, especially as the raw and meta outputs of challenges can be numerous and complex. Assigning sufficient resources for structuring and evaluating these outputs helps extract meaningful analysis, discussion, and conclusions.

A **post-challenge paper** represents one cornerstone of such activities. It conveys results, lessons, gaps, and research directions in a concise, intelligible form. Such a paper often keeps the community engaged by disseminating challenge outcomes, contributing new insights, and incentivizing future work to push the state of the art. Ideally, this post-challenge paper is complemented by:

- A white paper for a broader audience, explaining the challenge background,

- A challenge design paper describing the modelling and problem implementation,

- Code or dataset documentation for reproducibility and further experimentation.

## 1.2 Types of Post-Challenge Papers

In practice, many different types of post-challenge papers are possible, each providing distinctive benefits:

- A short analysis paper by organisers only (based on a fact sheet) comparing the best solutions' performance and reflecting on challenge design.

- A federated paper that includes top participants, with a stronger focus on best solution descriptions.

---

4. https://blog.neurips.cc/2024/06/04/neurips-2024-competitions-announced/

- An introduction to a book or special issue that collates multiple participant papers.

- An introduction to the proceedings of a workshop.

- A journal paper, possibly following several iterations of a competition series, providing a more in-depth analysis of how the challenge altered the scientific landscape and emphasizing the applicable current state of the art.

The exact nature of a post-challenge paper depends on the competition's objectives: addressing a fundamental AI/ML question, or tackling an applied problem, or introducing a novel formulation of an existing issue, or focusing on feasibility, benchmarking, and pushing the current state of the art.

When planning such a paper, organizers should consider the diversity of possible readers and carefully position the content. Different audiences may include:

- AI/ML researchers looking to apply their approaches to various problems (federated paper or introduction to workshop proceedings),

- AI/ML researchers wanting to learn about state-of-the-art methods (federated paper, introduction to special issue/book, or journal paper),

- Domain expert researchers interested in outcomes relevant to their field and comparisons with non-AI methods (short or journal paper),

- Domain expert scientists seeking models and problem framings that best attract the AI community (short paper),

- Challenge organizers searching for best practices and innovations in setting up and running competitions (short or journal paper),

- Scientists investigating evaluation frameworks (short or journal paper),

- Research managers looking for fruitful collaboration with skilled teams (federated paper),

- Vendors/investors seeking promising approaches in next-generation applications (short or journal paper),

- Science popularizers/reporters (short or journal paper).

Other additional aspects might also be of interest, such as considerations of ethics, diversity, AI for good, AI democratisation, framework development, and resource requirements.

The chapter guides readers through critical aspects of post-challenge planning: organizing raw outputs (Sec. 2), facilitating workshops (Sec. 3), drafting post-challenge papers (Sec 4), and establishing enduring benchmarks (Sec 5), concluding with actionable recommendations in (Sec. 6).

## 2 Challenge raw output

Having the above mentioned high-level considerations in mind and before describing typical post-challenge papers more concretely, we will now review the different kinds of challenge outputs that could be made available and of good use. The raw output of the challenge is all the material produced during the challenge, which must be analysed. The various types of outputs are listed in this section.

## 2.1 Competition platform output

A typical challenge platform can provide to the organisers, for each participant (or team) and all their submissions, many pieces of information: the time of submission, public and private scores and other similar quantities and most likely the detailed content of the submission, including the actual code in case of code submissions. The final (or selected) submissions are the most interesting, but intermediate ones can also inform the improvement process. Also, the time evolution of the public and private leaderboards is part of the challenge narrative.

## 2.2 Participants fact sheet

At the end of the competition, it is good practice for organisers to submit a form, the "fact sheet", to the participants to (i) have details on their best submission, which is not automatically provided by the platform, and (ii) know more about the participant themselves, who they are and how they managed their participation. It is best to advertise the form already in the last few days of the competition before participants move on to other activities and to insist that all inputs are worthwhile, even from non-top performers. The form should have a good balance between closed form Multiple Choice Questions, more straightforward to analyse, and free fields to gather specific feedback. The form should be anonymous by default, but it should also include the possibility of indicating the platform user name or leaving an email to continue the dialogue. The goal is to provide an overview of all participants (at least the ones who answer), while the bulk of the post-challenge activity will (and rightly so) focus on the best or most original contributions.

One might want to know more about:

- techniques and tools used

- resource used, in particular, training resources

- estimate time spent on the competition

- participant's background and initial knowledge about the competition. How diverse were the participants? From which age range? From which regions of the world? From academia or industry? Is there initial expertise on the problem or the domain? Which family of methods do they come from? Were they here to win, learn or find a dataset to use?

- How did the participants initially learn about the competition? This helps understand to whom the competition was eventually best addressed and disseminated and if that matches what was expected.

- Which available material and resources (papers, documentation, tutorials, baselines, tools, compute credits) did participants know of, and how useful and easy to use were they? This could help understand if the main competition features were understood or if any were missed, if that helps participants stay active and engaged, and if everything was eventually there to help them learn and participate. This would help organisers improve those resources for future benchmark or competition iterations.

- How interesting and challenging did participants find the competition and its format, with any pros and cons? What does the entry cost and learning curve look like for most participants? How resource-intensive was the competition to reach competitive performance? How difficult

the competition eventually was? These can explain participant activity during competition. This should bring some lessons on the competition formulation and calibration. You can eventually ask them how satisfied they were with the competition, if they'd want to stay engaged and how.

- Other questions can consider organisational aspects during the competition, such as ease of competition platform and submission protocol, competition length and phases, communication clarity, forum activity, prize distribution and incentives for participant investment, and opportunities for collaboration among participants.

This form and questions will help better assess which aspects the competition was most successful at and could be improved later.

## 2.3 Code

In a code-less platform, participants train their algorithm on a training dataset, apply it (the inference part) to a test dataset and submit the results as a data file. In this case, asking participants to submit their software, including training and inference, is customary on a platform like GitHub. This, however, can only be made mandatory for participants who can claim a price, which should have been specified in the competition rules; the others would often not bother. Also, if the inference code can be tested relatively quickly, it is much less the case for the training code, as the final model is often the result of many iterations. The code should be runnable, well documented, and accompanied by a short document describing its functionality.

For a platform accepting code submissions, the code (guaranteed to be the one producing the ranked results) is already available. There still needs to be a submission of commented human-readable code accompanied by a short document. It is the same if the code submission is only the inference part; the training part must be submitted separately.

There is a difference between publicising the code and releasing it to the organisers. The former is best for dissemination, but people in some communities might prefer to avoid it. It is also important to recommend that the code has an open-source license so that others using it don't risk copyright infringement.

## 2.4 Competition log

When the challenge is running, a competition logbook should be updated with the main events so that the narrative of the challenge can be told afterwards. Possible salient events: significant changes in the leaderboard, popular posts in the forum, for example, advertising a technique that many participants adopt, updates in challenge documentation, possible challenge reset after a flaw was uncovered and fixed, reports or the discovery of cheating, social media visibility, media coverage (and impact on participation), etc.

## 2.5 Unorganized raw output

A flow of information from the competition not formatted by the platform or online forms needs to be analysed. They can also represent a measure of how active the competition is. The competition forum is the primary source of such information: data exploration posts and notebooks, insights, code sharing, and documentation gleaned on the web on the topic. One can, for example, see

an idea appear in a forum post, followed by a code implementation (not necessarily by the same person), followed by a general increase in the scores on the leaderboard. However, relevant technical discussions may happen outside the competition forum, such as on social networks, blog posts, or arXiv papers. Also, competitions are often used as practical projects for (e.g., data science) courses. Compared to typical participants, students are usually compelled to write a hopefully clear document about their techniques, which might be public. Such spontaneous output should be harvested regularly, which is much easier if the competition has a unique acronym to be googled and for which alerts can be set up.

## 3 Post-challenge workshops and discussion

One or more post-challenge workshops are often organised as part of a conference or in a dedicated venue[5][6][7]. Participating in it is part of the incentive for participants. It allows participants to switch from competition to cooperation. They can discuss their techniques between them and with the application or Machine Learning experts. The connections created during such workshops encourage participants to remain engaged with the competition topic instead of just moving on to a different one. The discussions during the workshops and continued online are crucial to deriving the scientific lessons from the workshops and the avenues for further study and possible future challenges.

## 4 Post challenge paper template

This section details a complete template of a typical post-challenge paper. It should only be considered as guidelines. The emphasis on the different sections would vary widely depending on the type of challenge. Also, a decision to be made early on is how to include the authors of the most interesting contributions: should they write a sub-section about their methods (as done in this template) and sign the paper, or should they write their own paper citing the post-challenge paper? Examples of graphs relevant to a post-challenge paper illustrate the template.

### 4.1 Introduction

As for every paper, a proper introduction should first recall the context of the challenge and highlight the problem at hand. It should further explain why it is essential to solve it, how impactful it can be, and what the bottlenecks to addressing it have been so far. Eventually, objectives and expectations for such a challenge could be shared.

### 4.2 Challenge Description

This section should ideally come as a reminder and synthesis of a prior paper on the challenge design. One should first review the problem of the challenge to give a good enough understanding to the targeted audience for the remainder of the paper, particularly highlighting its main dimensions of complexity and variations. One should further present the specific task formulated for the challenge with some description of the underlying datasets, framework (such as a chatbot or reinforcement learning framework if relevant) and problem modelisation (possibly relying on some simulator).

---

5. https://autodl.chalearn.org/neurips2019
6. https://llm-efficiency-challenge.github.io/schedule
7. https://fair-universe.lbl.gov/

One can remind if the challenge aims to deliver a new problem or formulation to the Machine Learning community or advance and benchmark state-of-the-art. A section on related works and similar challenges can be written to best position this challenge in the scientific landscape and possibly build upon previous work.

The scoring metric and protocol should be discussed, and some considerations should be shared as to why they were chosen a priori to evaluate any advances towards problem-solving best. A brief description of the challenge platform , as well as possible specific choices (like resource allocation) or developments that can be justified there.

Finally, one can describe the challenge organisation and materials available to the participants, highlighting any innovations to increase participant engagement during the competition.

### 4.3 Challenge Narrative

Once the competition is over, it is interesting to understand retrospectively what happened during the competition, leading to the final leaderboard and results. Was the competition tight or not? How long did it take participants to reach a good enough performance? How many teams showed sustained activity, and how many eventually performed well? Did any innovation in solutions occur within the competition, breaking some performance ceiling? Or were most of the solutions derived from an adopted model published in the baselines or by a participant? These can be extracted from raw competition output, highlighting the competition dynamics, attractivity and difficulty.

For example, Fig. 1 shows the progress of participants in one competition as a function of time; one can see the bulk of participants progressing as a swarm, following community understanding of the problems, while a few isolated outliers obtained the best scores.

Fig. 2 shows participants' progress in a different competition in the (accuracy, speed) plan. Various strategies are evident: some tried to optimise both simultaneously, why some others (the best), `fastrack` and `gorbuno` have first reached the best accuracy, then improved their speed without accuracy loss.

A graph like Fig. 3 can help summarise the main events and competition activity.

### 4.4 Post challenge checks

In most cases, the performance of submissions can be evaluated against a held-out (or private) dataset, providing the final ranking. Generally, this is automatically done on the challenge platform, but it may require some manual actions, mainly when doing it automatically requires too many resources. The stability of the ranking can then be evaluated as in Fig. 4. The graph shows that the rankings among teams are stable, except for `4th Rek`, which performs significantly worse in the second phase.

A more detailed analysis is also possible, as in Fig. 5. The comparison of the private curves shows that a numerical analysis confirms that `Gabor` (the winner) is clearly above the others. The comparison of private and public curves of different participants shows a clear overfitting case for `Lubozs`, who was first on the final public leaderboard but slipped to seventh position in the private leaderboard. It turned out that he had indicated in his blog that he had set up an automatic cron job, which was automatically re-submitting new submissions (five times per day, which was the maximum allowed by the platform) with slightly altered parameters to maximise his public score. This engineering feat allowed him to select a lucky spike and grab the top of the public leaderboard, but this did not fool the private leaderboard measurement. It should be noted that this

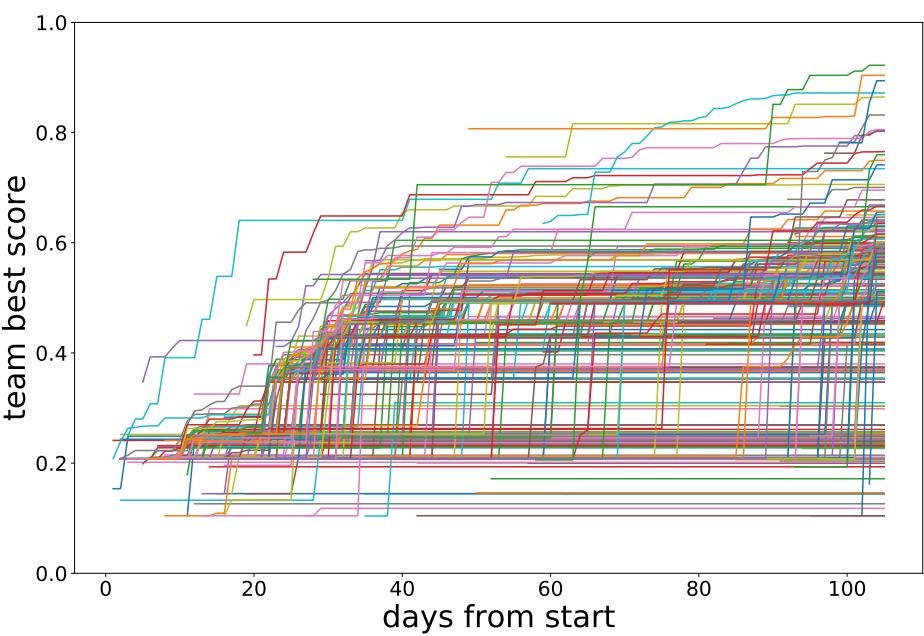

Figure 1: Participants' best score evolution as a function of days in the competition (Amrouche et al., 2019).

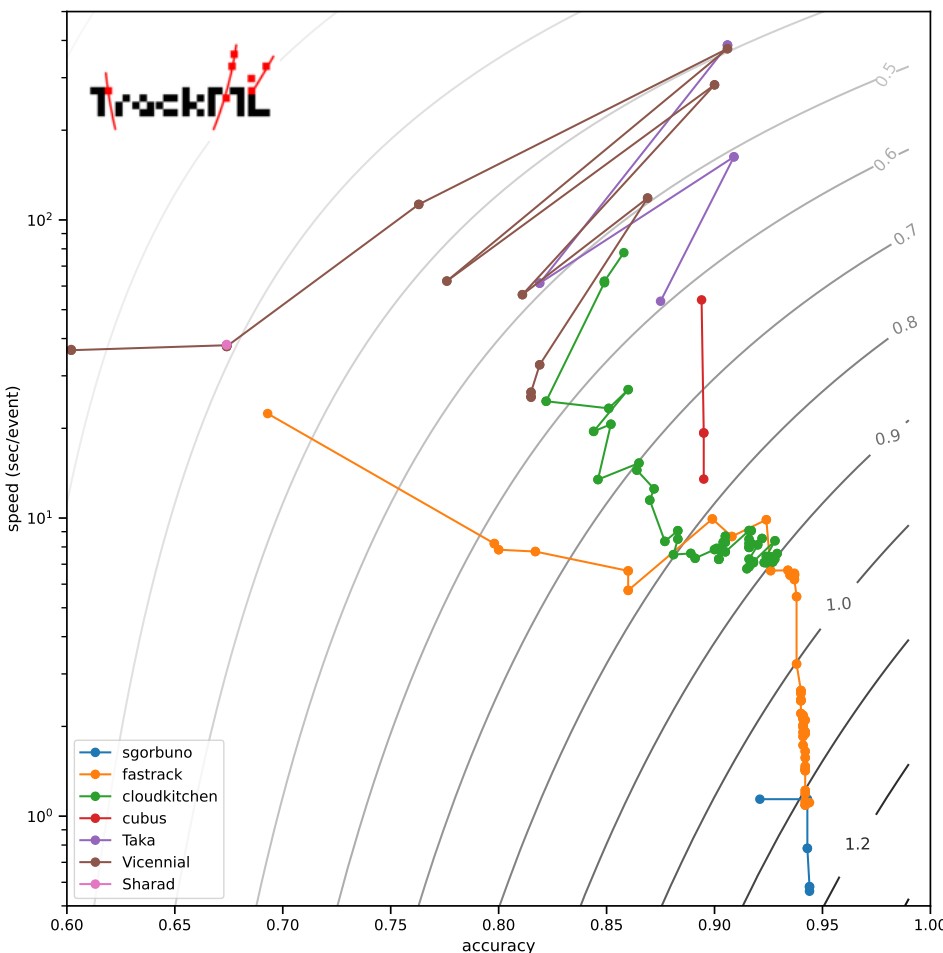

Figure 2: Participants score evolution: the horizontal axis is the accuracy, and the vertical axis is the inference speed. The total score, a function of both variables, is displayed in grey contours. Each colour/marker type corresponds to a contributor; the lines help to follow the score evolution(Amrouche et al., 2021).

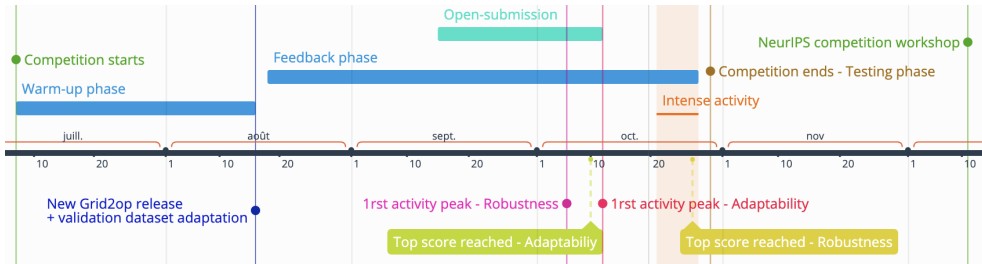

Figure 3: A high-level retrospective competition period timeline can help understand the competition organisation through phases and noteworthy events such as competition adjustments, peak of activities, performance outbreaks, and collaborative periods. This can support the competition narrative. (Marot et al., 2021)

piece of information was not reported through the standard channels or forum; it was discovered by googling, illustrating the point of analysing unorganised information as discussed in Sec. 2.5.

Additional graphs can be produced from by-products of solution submission gathered by the platform. For example, Fig. 6 compares the performance on different datasets, giving for each the intrinsic and modelling difficulties. Ideally, organisers should choose datasets of low intrinsic difficulty and high modelling difficulty, which would be in this example `Isaac2`, `Caucase` or `freddy`.

One can also analyse the solutions to evaluate their originality. For example, Fig 7 shows the dendrogram of a clustering competition. The dendrogram shows how similar or not the clusters found are, which correlates well with what is known on the participant's method or background: the set of six participants at the top of the diagram (#12 to #9) have highly optimised the starting kit using generic clustering algorithms; #3 and #4 are domain experts with the same background, who have found similar clusters with optimised domain methods; #1 is a CS student who has spent a lot of time studying the domain literature, has seen similar clusters as #3 and #4; on the other hand, the diagram sets apart #2 who is a CS expert who has developed a very original approach (which turned out to be impractical because of the significant resource it requires).

## 4.5 Deeper analysis of the submissions

After a challenge finishes, we often need a systematic and deep analysis of the winning solutions. The analysis could be very case-specific depending on the challenge task and application. As a consequence, we only mention a few general analyses here.

**Reproducibility** As an essential aspect of machine learning, reproducibility often should be considered in challenge organisation. Indeed, in post-challenge analysis, we should first reproduce winning methods to have a sanity check when the code was not submitted to the platform. The training should be reproduced even if it can be in practice challenging.

**Metrics.** In a challenge, organisers often use a single metric to evaluate people's submissions. The choice of the metric might come from organisers' interests, but we often need additional metrics to fully evaluate, understand and conclude with winning solutions. For example, in a classification task, we may need accuracy and balanced accuracy to pay attention to the skewness of dataset classes; in a regression task, we may need to mean squared error (MSE) and mean absolute error

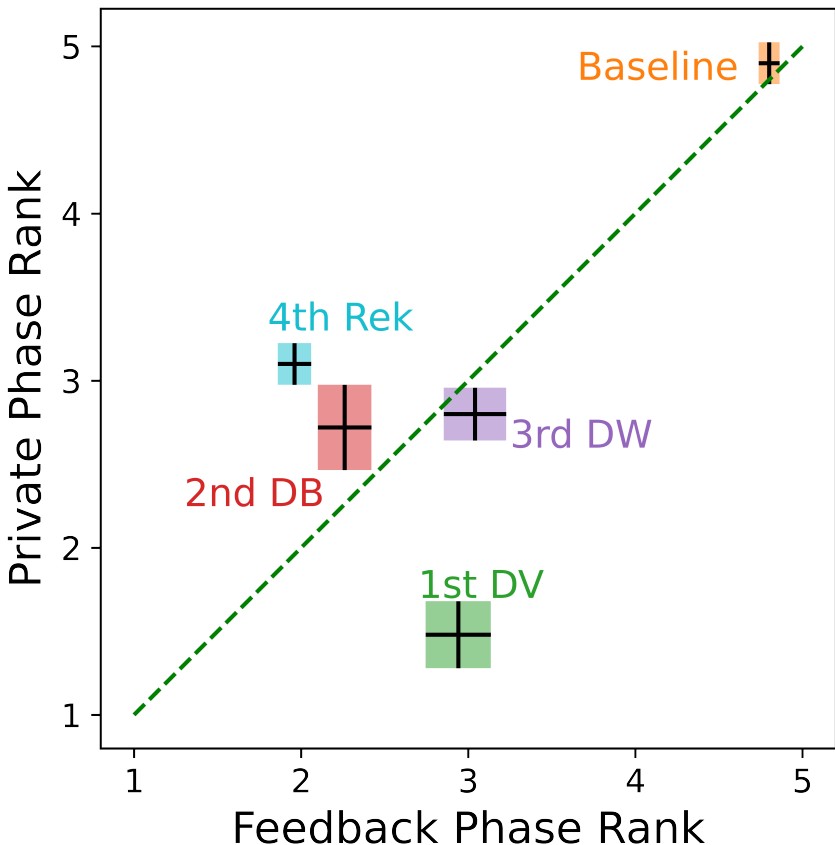

Figure 4: Overfitting/Ranking stability plot from (Xu et al., 2021). The comparison of the x/y axes shows overfitting. In a challenge of two phases, the feedback (or public) phase is the first phase, and the private phase is the second. By showing how submissions perform in the consequent two phases, we demonstrate the overfitting of algorithms. The rectangle shows ranking stability around each team. This rectangle is calculated by the average ranks of multiple reproductions of submissions' performances.

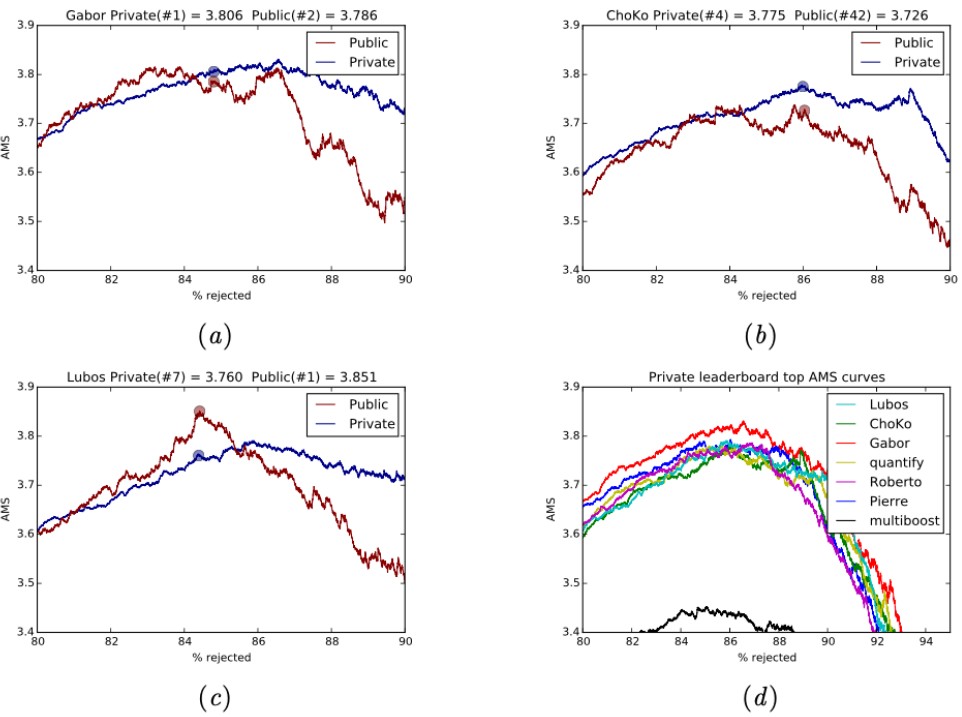

Figure 5: AMS Significance, a figure of merit of a classifier in a physics particle discovery context, as a function of decision threshold for different submissions in the HiggsML challenge (Adam-Bourdarios et al., 2014). The submission score is the maximum of the curve. (d) shows the private curves for different participants. (a),(b) and (c) compare the public and private curves from three participants, `Gabor`, `Choko` and `Lubozs`, with dots indicating their maximum values. The private curves are smoother because they are evaluated using more examples.

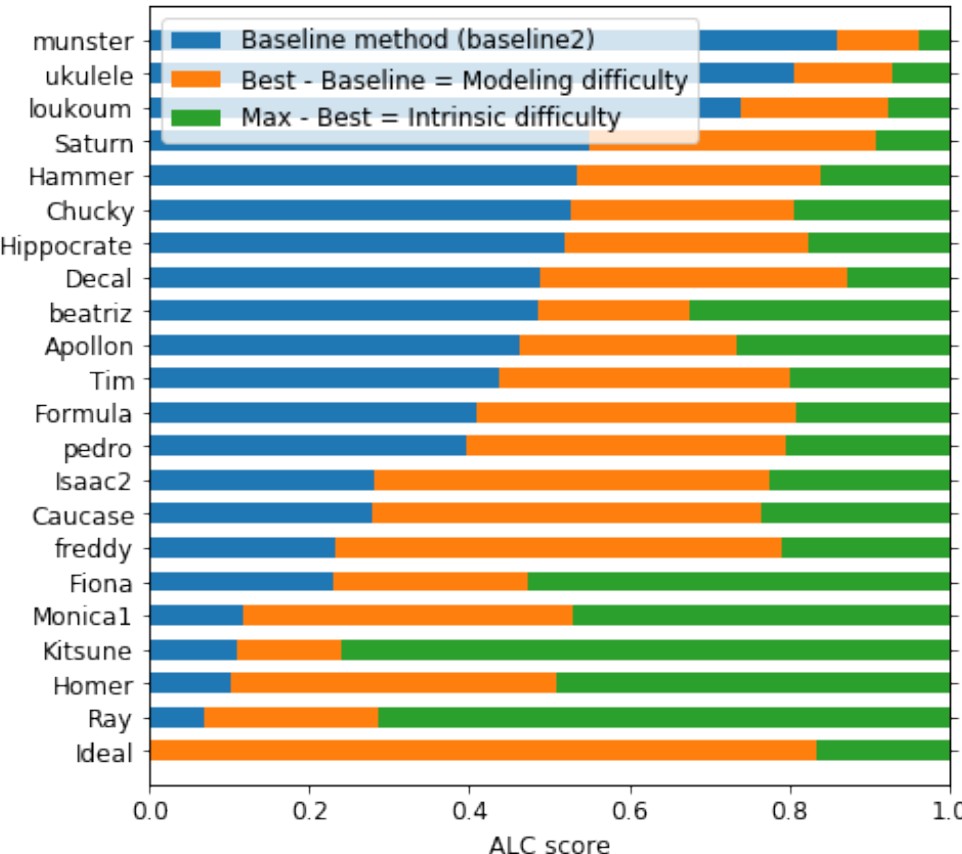

Figure 6: Measurement of task difficulty from (Liu et al., 2020). Each row corresponds to one dataset. Two difficulties are calculated here: intrinsic difficulty (maximum score minus best participant's score), shown with the green bar and modelling difficulty (best participant's score minus baseline score), shown with the orange bar.

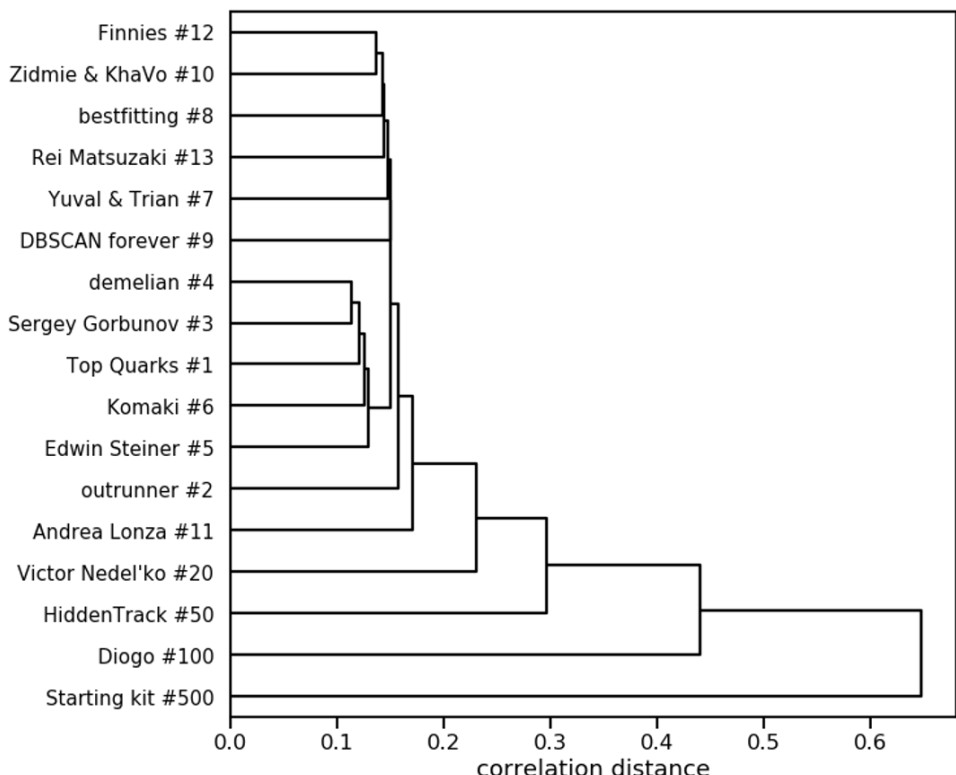

Figure 7: Dendrogram of the best solutions submitted by participants (name and final rank indicated) to the TrackML challenge (courtesy of authors of (Amrouche et al., 2019)). The diagram shows the thirteen best participants, plus participants ranked 20th, 50th, 100th, as well as the starting kit.

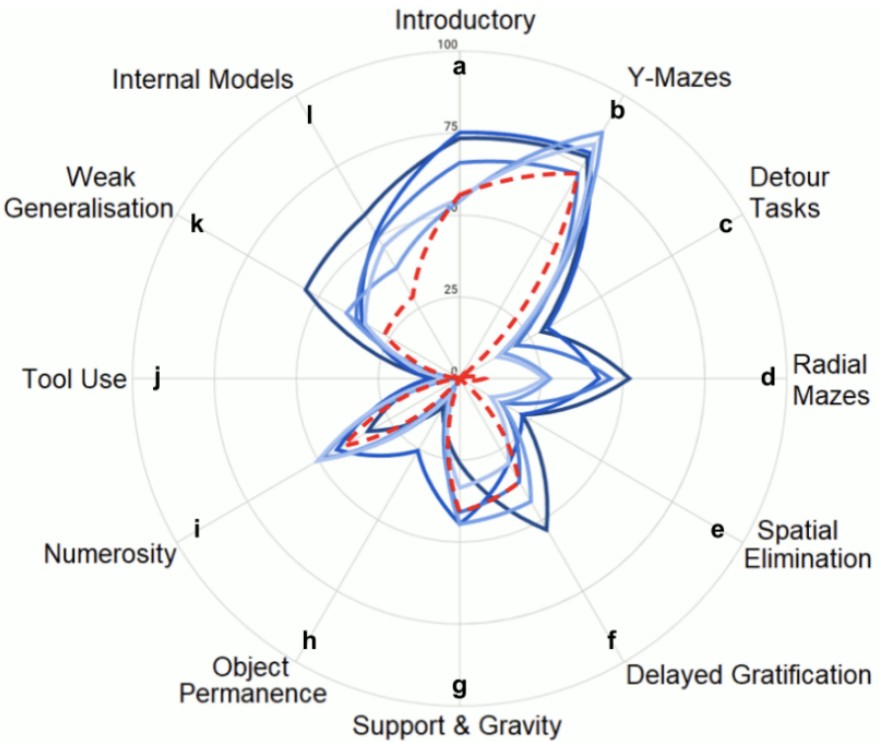

Figure 8: When a competition involves different tasks or different underlying metrics, a radar plot can provide a nice visual comparison of the submissions along all those dimensions as (Crosby et al., 2019) which tested agents on several tasks.

(MAE) for relative and absolute error analysis. Even though the organisers chose a fused version of multiple metrics, we could also evaluate each error component of the metric individually to understand the differences made by different methods.

For example, Fig. 8 shows the performance of different submissions for different tasks.

Another example, in Fig. 9 shows the performance of some of the best participants in a clustering competition (same participants as in Fig. 7). Although the participants had to optimise a single score, domain experts were satisfied to see these graphs showing that the best submissions maximise their cluster-finding algorithm's robustness (concerning ground truth parameters). One notable exception is #100, the only one showing a rising contribution in the bottom left graph. It indicated it was accidentally optimised for abnormal clusters, yielding a poor overall score, which was still interesting to domain experts.

**Pipeline.** A challenge solution usually consists of a pipeline of steps, e.g., data preprocessing, feature engineering, model training, hyperparameter selection, ensemble, etc. It is super interesting to investigate step by step the choices available from participants and ablation study the contribution of options. Such a study is not trivial because we need to split the steps of solutions, which is not necessarily logically clear; secondly, we need to modify many solutions to evaluate the options.

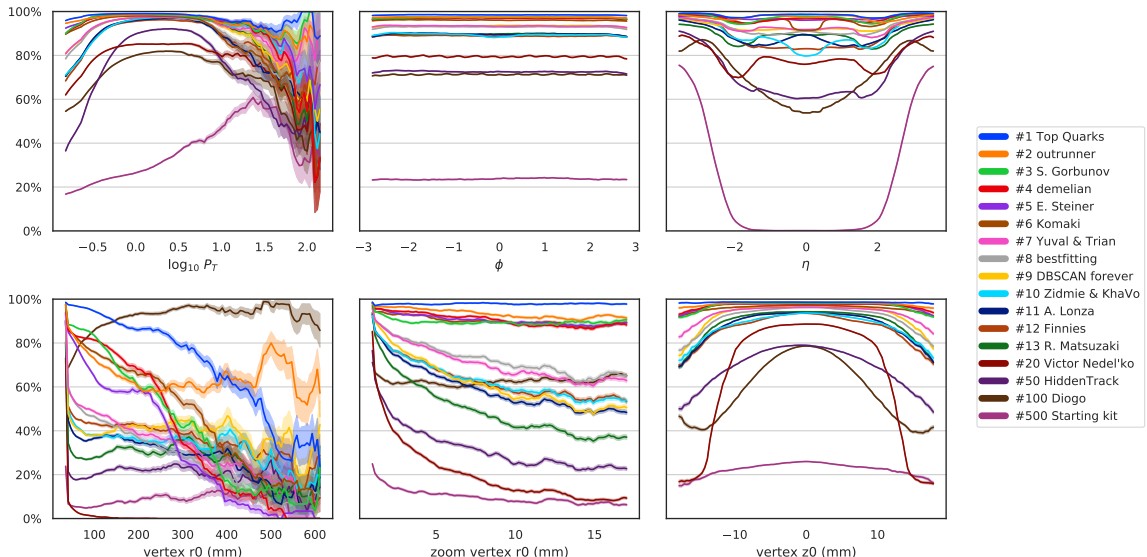

Figure 9: Cluster finding probability as a function of six ground truth cluster 3D shape and dimension parameters, from the TrackML Accuracy competition (Amrouche et al., 2019)

Once an ablation study is done, we could further ensemble the best options and output a revised version.

**Ensembling.** For each submission, the participant made a decision for each pipeline step. For some challenge types (i.e. classification), it should be possible to automatically ensemble different submissions to see if something could be gained from combining them (Kégl et al., 2018). In general, human ensembling (mixing and matching the various decisions taken at each step) can bring insights and improvements to even the very best submission. For example, one participant has built clever features but could have used them better.

**Generalizability.** In addition to the setting used in a challenge, we could also look at the generalizability of methods under different settings, including different datasets, time resource constraints, memory scalability, etc. This concept of generalizability has already been taken into account in AutoML challenges. For non-AutoML challenges, it is thus interesting to investigate.

Although these analyses are time-consuming, they can be very fruitful and worth the effort; they should be seen as the final processing step of the challenge's fruits (the submissions). They could be the theme of post-challenge workshops.

## 4.6 Description of the most interesting submissions

If one paper is written, which hosts contributions from the most interesting submissions (not necessarily the absolute best but also the ones deemed original), this section would hold sub-sections, each describing a submission. Participants should write these sub-sections themselves, following guidelines or a template provided by the organisers. Otherwise, the organisers can write themselves one or more subsections based on the material available to them.

If the participants have written separate papers, only summaries would be necessary here.

## 4.7 Scientific outcome

This section summarises the scientific outcome of the challenge based on the deeper analysis of the submissions and the individual submission descriptions.

- What new insights were brought by the challenge for data science and the domain?

- What techniques worked best in the different pipeline steps?

- What about the explainability of these techniques?

- What are the hardware/resource constraints?

- Are these originals in absolute or in this particular domain?

- What techniques did not work?

- What are the new avenues for further studies?

## 4.8 Lessons on the challenge organisation itself

The scientific outcome is the primary lesson from a challenge. However, another important one is the feedback on the organisation of the challenge itself.

Answers to the following questions should be sought:

- Did the participants solve the problem they were supposed to solve?

- Any fundamental flaw in the competition?

- Was the metric appropriate? How could the metric be improved?

- Was the dataset suitable? How could it be improved?

- Were the tasks of the challenge of a difficulty adapted to push the state-of-the-art in the domain considered?

- Feedback on the platform and the challenge mechanism.

- Some measurement of the popularity of the competition and comments on advertisement and dissemination.

- Participants sociology diversity (from fact sheet)

## 4.9 Conclusion

The paper's conclusion would summarise the scientific findings and sketch possible future actions, permanent datasets and benchmarks or future challenges.

Table 1: Comparisons of competition and benchmark (Xu et al., 2022).

|  | Competition | Benchmark |
|---|---|---|
| Purpose | Crowdsourcing problems in a short time and harvesting solutions | Continuous fair evaluation over a long period, in a unified framework |
| Phases | Multiple phases | Single-phase |
| Time period | Usually limited | Often never-ending |
| Cooperation & information sharing | Limited due to the competitive nature | As extensive as possible |
| Submissions | Usually algorithm predictions or algorithm code | Algorithm code or datasets; code or dataset name, description, documentation meta-data and fact-sheets; scoring programs for custom analyses |
| Outcome | leaderboard with usually a single global ranking based on one score from each team (last or best) | Table with all the submissions made; sorting with multiple scores possible; multiple analyses, graphs, figures, code sharing |

## 5 Post challenge benchmark

Benchmarks differ from competitions in many ways, as summarised in Table 1. We organise a competition to crowdsource a task and harvest the winning solutions. This competition usually lasts a couple of months and has multiple phases (public, private, etc.). We intentionally rank participants linearly due to their competitive nature, and people are not allowed to share the code directly. However, for benchmarks, we are interested in a research task and would like to invite people worldwide to contribute continuously. The benchmarks last much longer than competitions, usually never-ending, and only one phase is associated. Another big difference is that benchmarks encourage people to share ideas, solutions, code, and findings as much as possible because the goal is to push forward this research task. Thus, rich publications, seminars, and workshops are expected for communication.

By turning a challenge into a benchmark, we gain multiple benefits for different people. Challenge organisers give people around the world more time (possibly never-ending) to join the benchmark and make submissions to push forward the research task. Participants have more time and possibly can open-source datasets for their own research, and the leaderboard of research gives credit to their methods. For the platform of benchmarks, it is always better to have high-quality benchmarks and attract more people.

Turning a challenge into a benchmark is usually labour-intensive and repetitive. We hereby develop the codabench project to host benchmarks easily and free of charge. Technically, organisers only need to prepare data, logistic code for digesting and evaluating, and a configuration file. Benchmarks will be run in separate dockers; thus, the results will be reproducible.

## 6 Conclusion

In the chapter, we have covered the main actions in favour of the long-lasting impact of a challenge, particularly with a post-challenge paper template and advice on how to turn the challenge into a benchmark.

The main point is that significant time and person-power resources should be allocated up-front to this activity, which will harvest and make sense of the wealth of information produced by the challenge. For example, if the challenge is funded through a call, it would be best to foresee at least one year for the post-challenge activities.

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
