# OpenReview forum: "Towards impactful challenges: post-challenge paper, benchmarks and other dissemination actions"
_DMLR — Accepted by DMLR_

### Review · Reviewer_zASh · 2024-10-23

**Recommendation:** 1
**Confidence:** 3

**Summary Of Contributions:**

This paper concerns itself with activities around a data-driven machine learning competitions as organized by NeurIPS, kaggle and so on.
Based on a definition of the goals such competitions are claimed to have and perceived participants, it contains a wealth of advice and ideas for competition organizers.

**Strengths:**

- The paper is clearly written by people who have run competitions and hence the advice provided is explicit, actionable and of value.
- Section 5 is well written and contains a wealth of information.

**Audience:**

Yes

**Broader Impact Concerns:**

I have no concern about the ethical implications of this paper and no broader impact statement is warranted.

**Claims And Evidence:**

The paper makes some prominent claims (stating the purpose of competitions in general, identifying stakeholders) but provides little to no evidence or analysis.

**Datasets And Benchmarks:**

not applicable here

**Extended Submissions:**

not applicable here.

**Limitations:**

- The paper stipulates some definitions which are hard to reconcile with reality. For instance, it holds that the main purpose for a competition is "to address an open problem and engage a whole community towards solving it, not only at one point in time but hopefully in the long run".
Although this is true, for sure, for some competitions, this does not apply to many kaggle competitions where the data provider is more interested in solving a one off problem well enough for real-world application. The scientific merit is then either a novel method or a mash-up of known methods that provide a good enough solution that can be reapplied to similar problems. In fact, many participants "learn" to compete over several competitions and build up an arsenal of solution approaches they repeat. This also means for the paper that the black and white view of competition aftercare is maybe somewhat restrictive. A competition that engages the community to solve a problem once well enough and then highlights, say, that solution approach for future generations makes a valuable scientific and societal contribution and will not necessitate any future benchmarks. At best, it leads the way to a tougher new challenge which stands on its shoulders.
- The paper identifies 4 key stakeholders in each competition: organizers, domain experts, AI/ML experts in techniques relevant to the problem posed and participants. This definition is curious, in several aspects. Firstly, it ignores one large stakeholder: the data provider (simulation provider for RL). Data is the life-blood of competitions and owners of large datasets, be it commercial entities, academic institutions, government sponsored entities, ... need their needs met. Secondly, some competitions exist, because there are actually no AI/ML experts in techniques for such datasets yet. In fact, open competitions are attractive to data providers because they bring new insights by communal engagement.
- Competitions will not work without participants' engagement, which in turn depends on participants' motivation which then should be dead center of the competition design, whose outcome needs to satisfy the data providers/organizers' aims. Squaring that circle hard and is curiously unaddressed in the paper. For instance, many competitions have monetary prizes for successful participants. That in turn means, that competitions must be designed so that outcomes can be ranked (even legally, otherwise participants can sue for prize money) which in turn make pursuing some valid non-rankable scientific questions like "find a better evaluation metric" hard to pursue in that mode. The paper leaves it unaddressed whether there are competitions which attract participants more for monetary reasons or for the prestige (even then you still need ranking) or for a qualitative reason (like some competitions in CAMDA, which seemingly attract fewer participants). Not recognizing these real world challenges means the paper seems to not apply to many real world competitions. For instance, many real world challenges are organized in competition series, where the quantitative and qualitative learnings of the last competition is taken into account and incorporated into the new challenge whilst keeping the franchise and past participants engaged - a common modus operandi not considered in the paper.
- It would be great if the paper contained more examples of real world competitions to underline its findings and claims, as well as basic statistical evidence for some of its more prominent claims, like the ones highlighted in the first point .

**Requested Changes:**

Apart form section 5, the language could be clearer. For instance "different paper kinds" probably should read "different types of papers", etc.

**Strengths And Weaknesses:**

The paper contains some concrete, actionable and hence valuable advice for competition organizers when thinking about
the long term impact of their competition at the design phase (see strengths below).
Nevertheless, the paper has some surprising weaknesses in the way it defines the general purpose of challenges as well as in its identification of stakeholders. This results in the paper not applying in the generality it might wish to.

---

### Review · Reviewer_rvDQ · 2024-10-23

**Recommendation:** 3
**Confidence:** 2

**Summary Of Contributions:**

This paper proposes guidelines for what organizers should do after AI/ML challenges. The main focus is on the post-challenge paper that performs a sort of a meta-analysis of different solutions along different metrics, analyzing difference and similarities, identifying room for improvement in the challenge platform, etc.

**Strengths:**

1. The topic is important and a good fit to the journal. I expect this to be of interest to at least some of the readers of DMLR.
2. Section 5 was detailed and had good examples. I enjoyed the figures reproduced from different post-challenge papers.

**The topic is a good fit, so I am willing to change my recommendation to accept as long as the authors sufficiently address my questions after a major revision.**

**Audience:**

Yes

**Claims And Evidence:**

No. See requested changes.

**Datasets And Benchmarks:**

N/A

**Extended Submissions:**

N/A

**Requested Changes:**

1. The biggest weakness of this paper at this stage is that the claims or proposed guidelines are not well-justified. Sections 1-4 are long lists of thoughts and opinions and not fit for a scholarly article at this point. There are no citations or even specific examples, making this read like a blog post.

Some concrete suggestions the authors may consider:

1-1. (Necessary for acceptance)  Sections 1-2 should be merged into one introduction section.

1-2. (Necessary for acceptance)  The new introduction section should motivate why readers should care about this topic. Are there any statistics on the number of AI/ML challenges or competitions? Are they on the rise? Are more ML/data science conferences establishing official competition tracks? Are there any blog posts from conference organizers that explain their rationale behind hosting more challenges? This information can be manually gathered from past and present conference websites.

1-3. Are there any statistics on post-challenge or post-competition papers? Can we connect it to numbers from above to make claims about the percentage of organizers who end up writing these papers?

1-4. (Necessary for acceptance) Current section 2 has a long list of types of post-challenge papers, which can easily be summarized into a table. More importantly, it needs citations of existing papers that fall under the different categories, even if they are not peer-reviewed publications. Otherwise, there is no justification behind why the authors chose these categories.

1-5. (Necessary for acceptance) Current section 3: Similar comments as above regarding citations. Are there competition websites that the authors think did a good job keeping a competition log, organizing git repo, or asking for feedback? There is also a ton of literature from social sciences about survey design, which could be cited in Sec. 3.2.

If the authors decide not to do this additional analysis, then they can instead significantly shorten this section and thereby reduce the claims they are making.

1-6. (Necessary for acceptance) Current section 4: Again, are there examples of this?

2. (Necessary for acceptance) Section 5 is the strongest section of this paper. However, the content is only coming from a few competitions (8 according to the references) and they are all dated at this point, with the most recent one being from 2020~2021. Considering the number of AI/ML challenges that happen every year, this is pretty thin. I suggest the authors look into more examples to establish a pattern or generalize their findings.

3.  (Necessary for acceptance) Section 6 seems not fully developed. It needs a lot more specific examples like Section 5, or taken out. Also Section 6 falsely claims "We hereby develop the codabench project [...]" which is already a published project.

4. (Necessary for acceptance) Paper title doesn't fit the paper content. It should only be about post-challenge paper unless the authors add more content to the other dissemination actions and benchmarks in the revision.

5. Writing:

5-1. Abstract needs to be more specific.

5-2. "Sec.2 is an introduction [..]" doesn't match up.

5-3. Repeated use of the word "chapter" instead of paper or something more fitting.

5-4. Repeated use of the word "federated paper" which is not an established term. Perhaps the authors meant collaborative paper.

5-5. Many vague comments: "The former is best for dissemination, but people in some communities might prefer to avoid it." - such as?

5-6. Language not fit for scholarly article: "It is super interesting"

5-7. Proofreading: inconsistent capitalization, etc.

**Strengths And Weaknesses:**

See separate sections below.

---

### Review · Reviewer_3n6g · 2024-11-27

**Recommendation:** 3
**Confidence:** 2

**Summary Of Contributions:**

The manuscript "Towards impactful challenges: post-challenge paper, benchmarks and other dissemination actions" provides a valuable guide for ensuring the long-term impact of AI challenges. While the paper offers useful frameworks and practical insights, several areas can be improved to enhance its clarity, accessibility, and engagement. Below are specific suggestions:

**Strengths:**

Mentioned Above

**Audience:**

Yes

**Claims And Evidence:**

Convincing claims.

**Datasets And Benchmarks:**

Sufficient information

**Extended Submissions:**

None

**Limitations:**

Mentioned Above

**Requested Changes:**

Mentioned Above

**Strengths And Weaknesses:**

Abstract: First Sentence Improvement: The current opening sentence of the abstract could be more engaging and impactful. Suggestion: Replace "Organising an AI challenge does not end with the final event." with "The conclusion of an AI challenge is not the end of its lifecycle; ensuring a long-lasting impact requires meticulous post-challenge activities."

Abrupt Transition: The sentence "The target audience of different post-challenge activities is identified." feels abrupt. Suggestion: Rephrase to "This work identifies target audiences for post-challenge initiatives and outlines methods for collecting and organizing challenge outputs."

Introduction: Generic Description of Participants: The phrase "The participants are diverse; they can be experts from the application domain or AI, students or seasoned data scientists." lacks depth. Suggestion: Update to "Participants come from diverse backgrounds—ranging from domain-specific experts and AI specialists to students and experienced data scientists—each contributing unique perspectives that enrich the challenge outcomes."

Structural Preview: The introduction contains a dry summary of the chapter structure ("This chapter is organised as follows: Sec. 2 is an introduction, Sec. 3 details the raw output..."). While functional, this could be rephrased to make it more engaging. Suggestion: "The chapter guides readers through critical aspects of post-challenge planning: organizing raw outputs (Sec. 3), facilitating workshops (Sec. 4), drafting post-challenge papers (Sec. 5), and establishing enduring benchmarks (Sec. 6), concluding with actionable recommendations in Sec. 7."

Section 2: Purpose of Post-Challenge Activity: Repetitive Phrasing: The sentence "To keep such a community engaged and hopefully growing, it is important to share with them and the wider research community..." feels repetitive and could be streamlined. Suggestion: "Sustaining and growing such a community requires sharing outcomes and resources with both participants and the wider research community."

Bullet Points: The current list in this section could be formatted for clarity and flow. Suggestion: Use active phrasing in bullets to emphasize actions:

Document results and lessons to build upon. Highlight remaining gaps and essential research directions. Provide resources like benchmarks, solution codes, and tutorials to facilitate further research. Section 3: Challenge Raw Output: Overloaded Details: Subsections like "Participants Fact Sheet" provide exhaustive detail, some of which may overwhelm readers. Suggestion: Consolidate examples or move granular details (e.g., survey questions) to an appendix.

Clarity in Subheadings: Subheadings like "Unorganized Raw Output" lack clarity. Suggestion: Rename to "Unstructured Data Sources" for better readability.

Section 6: Post-Challenge Benchmark: Limited Depth: While the section introduces benchmarks effectively, it does not delve deeply into the practical challenges of transitioning from challenges to benchmarks. Suggestion: Expand on potential pitfalls (e.g., resource requirements, participant retention) and solutions for maintaining community engagement.

Table 1 Presentation: The comparison between challenges and benchmarks in Table 1 is valuable but could benefit from an introductory sentence to provide context. Suggestion: Add a brief paragraph before Table 1 to explain its purpose and relevance.

Overall Suggestions: Ethical Considerations: The manuscript briefly mentions ethics but does not discuss it in-depth. Given the relevance of ethical AI, this should be expanded to include topics such as fairness in competition metrics, data privacy, and environmental costs.

Figures and Visuals: Some figures (e.g., Fig. 8) are not adequately integrated into the narrative. Suggestion: Explicitly reference each figure in the text, explaining its relevance and insights.

Conclusion: The conclusion reiterates content but lacks actionable takeaways. Suggestion: Summarize key recommendations and outline next steps for challenge organizers, such as allocating resources for post-challenge activities and establishing reproducibility standards.